# Top ten research priorities for Essential Emergency and Critical Care: A modified Delphi process

Ryan Rhys Ellis[1], Wurry Ayuningtyas[2,3,4], Alexandra Wharton Smith[5], Martin Gerdin Wärnberg[4,6], Carl Otto Schell[4,7,8], The EECC Prioritisation Group[¶], Tim Baker[4,9,10,11]*

1 Department of Perioperative Medicine and Critical Care, The Royal London Hospital, London, United Kingdom, 2 Dr. Soetomo General Academic Hospital, Surabaya, Indonesia, 3 Faculty of Medicine, Universitas Airlangga, Surabaya, Indonesia, 4 Department of Global Public Health, Karolinska Institutet, Stockholm, Sweden, 5 Department of Global Health and Development, Faculty of Public Health and Policy, London School of Hygiene & Tropical Medicine, London, United Kingdom, 6 Perioperative Medicine and Intensive Care, Karolinska University Hospital, Solna, Sweden, 7 Centre for Clinical Research Sörmland, Uppsala University, Eskilstuna, Sweden, 8 Department of Medicine, Nyköpings Hospital, Region Sörmland, Nyköping, Sweden, 9 Department of Emergency Medicine, Muhimbili University of Health and Allied Sciences, Dar es Salaam, Tanzania, 10 Department of Clinical Research, London School of Hygiene and Tropical Medicine, United Kingdom, 11 Queen Mary University of London, London, United Kingdom

¶ Membership of The EECC Prioritisation Group is provided in the Acknowledgements.
* tim.baker@ki.se

## Abstract

Critical illness leads to millions of preventable deaths each year. Essential Emergency and Critical Care (EECC) is a pragmatic, globally relevant approach designed to address critical gaps in basic life-saving care. This study identified the top ten research priorities to guide the development of EECC over the next five years. At the first global EECC Research Conference in November 2024, 46 clinicians, researchers, and policymakers representing nine predominantly high-income countries participated in a structured priority-setting process. Participants formed four thematic groups focused on the current state, implementation, impact, and refinement of EECC. Throughout five sessions, groups developed sub-themes, generated research questions, and identified key priorities. An anonymous survey was used to rank questions within each theme, followed by a final selection of the top ten. Participants developed 28 research questions across four themes. The final top ten included four on EECC implementation, four on its current state, and two on impact. No top-ranked questions emerged from the refinement theme. The top ten EECC research priorities focused on estimating the burden of critical illness and on implementing programs effectively. Advancing these priorities through collaborative research is essential to strengthen health systems and improve outcomes for critically ill patients globally.

**Data availability statement:** All relevant data are within the paper and the provided supplementary material. The manuscript contains summary data on the participants. Further extensive background, where supplied, for participants is available in S1 File. The manuscript presents the key results regarding the shortlisted questions from each theme and the overall combined top 1o priorities. The quantitative data underlying this choice (with mean Likert scores as well as SD, median and IQR) are included in S1 Data.

**Funding:** The authors received no specific funding for this work, and no individual author received any grants or direct financial support for the study. The conference that generated this work was supported by a Karolinska Institutet Research Conference Support grant (FS-2024:0010). No authors received personal funding from this grant. The funders had no role in study design, data collection and analysis, decision to publish, or preparation of the manuscript.

**Competing interests:** I have read the journal's policy and the authors of this manuscript have the following competing interests: there are no financial competing interests to declare. Authors RE, AWS, COS, and TB are non-paid Board members of EECC Global (https://www.eeccglobal.org/).

## Introduction

Critical illness has been defined as "a state of ill health with vital organ dysfunction, a high risk of imminent death if care is not provided and the potential for reversibility", and critical care as "identification, monitoring, and treatment of patients with critical illness through the initial and sustained support of vital organ functions" [1]. Patients who are critically ill may be neglected in health systems due to a lack of recognition, resource allocation, prioritisation, and timely identification and treatment with essential life-saving interventions.

Critical illness affects an estimated 45 million adults globally each year, with inadequate access to critical care contributing to millions of preventable deaths [2]. In Africa, for instance, one in eight hospitalised patients is critically ill, and approximately 20% of these patients die within seven days [3]. This is not an isolated issue; similar burdens of critical illness have been documented across diverse geographic and economic contexts, affecting both adults and children [4]. Notably, the majority of critically ill patients are managed in general wards rather than specialised units [5]. In low-resource settings, the availability of even basic supportive care is severely limited, and most critically ill patients do not receive essential life-sustaining care [6].

Essential Emergency and Critical Care (EECC) has been developed as the most simple, effective treatments and actions to effectively and pragmatically address the burden of critical illness and move towards ending preventable critical care deaths. EECC is the care that all critically ill patients should receive in all hospitals worldwide, irrespective of their age, gender, diagnosis, or social status. In 2021, 269 global experts from over 50 countries with clinical experience across acute medical specialities took part in a consensus project to reach an agreement on the content of EECC [7], which would be feasible for all hospital wards and units. The agreed-upon content includes foundational clinical processes for timely recognition and treatment of critically ill patients, as well as identification of the required components for hospital readiness. EECC focuses on basic lifesaving clinical processes such as appropriate oxygen delivery, airway manoeuvres, IV fluid administration and communication. EECC does not address more advanced processes such as intubation, ventilation and renal replacement therapy. From this consensus, EECC has begun to be integrated into health systems, encouraging a horizontal approach that focuses on illness severity rather than speciality or diagnosis to improve care and outcomes for all critically ill patients [2]. Further EECC tools have also been developed, including facility assessment surveys and an EECC training course.

Although previous global critical care research agendas have identified broad themes, they have largely focused on intensive care settings and advanced technologies and therefore did not generate priorities tailored to the basic, life-saving care that most critically ill patients require and that EECC targets [8–10]. EECC represents a fundamentally different, system-wide approach centred on simple, feasible, and universally deliverable care for all critically ill patients, including the majority who are treated outside intensive care units. With increasing global uptake of EECC following the 2021 consensus definition [7], a timely, structured research prioritisation process was needed to guide evidence generation, support implementation, and ensure that

emerging research efforts align with EECC's unique scope and aims. In this paper, we report the output of a consensus-driven exercise to generate global EECC research priorities for the next five years.

## Methods

### Study design

From November 6th to 7th, 2024, the first global research conference on EECC was held at the Department of Global Public Health, Karolinska Institutet, Stockholm, Sweden, in collaboration with Muhimbili University of Health and Allied Sciences, Tanzania. A range of clinicians and researchers were invited to participate, selected based on their research interests and relevant clinical backgrounds in emergency, critical care and public health (backgrounds of expert panel members outlined in S1 File Expert Participants). A total of 46 participants attended and contributed to the research priority-setting exercise. There was minimal attrition across rounds, with 41 participants providing final responses to rank the top 10 questions. At the conference, participants were presented with an update on the current state of critical illness, critical care, and EECC research, including insights from previous studies from Africa, Asia, and Europe. A concurrent modified Delphi study was conducted, with participants engaged in structured, interactive group sessions to develop and prioritise EECC research questions.

### Participants

Participants were purposively assigned to one of four groups to achieve a balanced distribution of expertise across experience, gender, and regional background. Participants in each group established research priorities in five sequential sessions, each representing one of four designated themes developed by the study leads: i) The current state of EECC; ii) Implementing EECC; iii) The impact of EECC; and iv) Refining EECC. Within each themed group, the members identified the prioritised research questions. The opinions of the gathered experts were gathered via a structured process, described by Schmidt [11], which involved multiple rounds of deliberation leading to a final consensus. This stepwise process progressed through issue generation, preliminary prioritisation, and final ranking via five defined sessions.

### Prioritisation process

The five sessions were conducted over two days with specific objectives to be achieved before progression to the next session. Within each session, a facilitator was assigned to each group to guide the respective thematic areas, and a note-taker was assigned to present the discussions and upload notes and decisions to an online document for analysis. A detailed description of this process is presented below and summarised below (Fig 1).

**Session 1.** Each of the four groups was tasked with generating research sub-themes within their designated theme.

**Session 2.** The groups reconvened to present the sub-themes to all conference participants. An anonymous digital poll was conducted using Menti [12] on participants' phones or laptops, allowing participants to propose additional relevant sub-themes. The combined input from Sessions 1 and 2 produced an exhaustive list of research sub-themes.

**Session 3.** Participants returned to their groups and developed research questions based on the sub-themes.

**Session 4.** Each thematic group was asked to shortlist seven research questions for the final ranking stage. This number was selected pragmatically through discussion among the study leads, balancing the need to capture sufficient breadth within each theme with the need to keep the final ranking exercise feasible for participants. Shortlisting an equal number of questions from each theme ensured balanced representation across thematic areas and prevented any single theme from disproportionately influencing the final priorities. During the conference, participants agreed that seven questions per theme provided a manageable number for meaningful discussion and comparison within the available time. Via consensus, each group selected the seven most important research questions they had developed, resulting in a total of 28 priority research questions across the four themes.

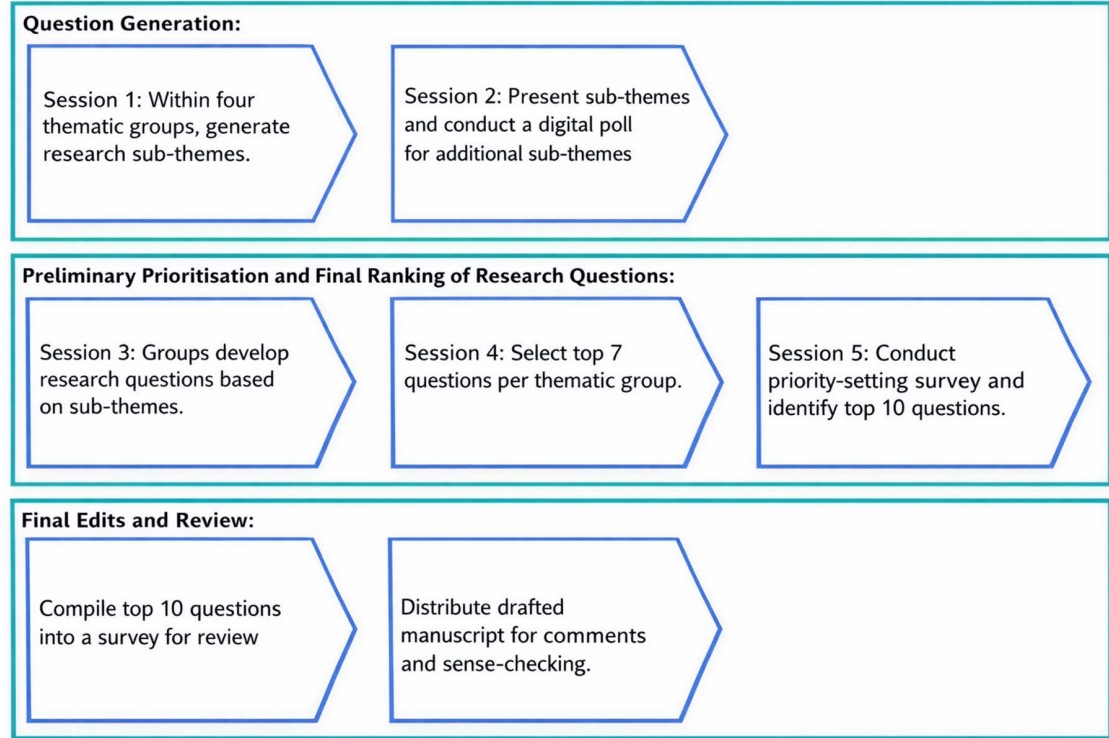

**Fig 1. Summary of the priority-setting process.**

**Session 5.** All conference participants completed a priority-setting survey using the online platform SmartSurvey [13]. The Smartsurvey software was used because the lead researcher for this section was familiar with it. There were no technical issues during the use of this software. The 28 questions were collectively reviewed, and participants rated each using a 5-point Likert scale ranging from 0 (not important at all) to 4 (essential). The top ten EECC priority questions were identified as those with the highest mean scores across all responses.

## Final edits and review

The top ten generated questions were compiled in SmartSurvey and distributed to the study leads for review, informing the drafting of the manuscript. The drafted manuscript was distributed to all conference participants for further comments and sense-checking.

## Ethics

This study involved only professional participants (clinicians, researchers, and policymakers) contributing their expertise during a structured research priority-setting exercise. No patient data or sensitive personal information was collected. Given the non-clinical and low-risk nature of the activity, formal ethical review was not required. All participants provided informed written consent, and survey responses were collected anonymously. Prior consensus-based research priority-setting projects involving experts and without direct patient data have similarly proceeded without formal ethics committee review [14,15].

## Results

Of the 46 participants in the prioritisation process (Table 1), just over half were female (57%, n = 26). Physicians constituted the largest occupational group, representing 65% (n = 30) of participants, followed by policymakers (7%, n = 3), nurses (4%, n = 2), and technical advisors (4%, n = 2). The majority had research experience, with 20% (n = 9) identifying exclusively as researchers. Participants brought a mix of experience from high-income countries (HICs) and low- and middle-income countries (LMICs). At the time of the conference, 50% (n = 23) were based in HICs, 17% (n = 8) in LMICs, and 33% (n = 15) worked across both settings. Nationalities represented included the United States, Nigeria, Tanzania, Ethiopia, Sweden, Norway, Indonesia, the United Kingdom, and New Zealand.

The seven priority questions for each of the four themes from Session 4 are presented in Table 2. The final ten prioritised research questions consisted of four questions from the theme of Implementing EECC, four from the Current State of EECC, and two from the Impact of EECC. No questions were chosen from the theme of Refining EECC (Table 3). The quantitative data collected and used for ranking are presented in S1 Data.

## Discussion

Ten prioritised EECC research questions have been developed in this prioritisation process held at the first global EECC research conference. The questions cover the current state of EECC, its implementation, and the impact of the EECC provision.

The questions in the current state of EECC focus on the global burden of critical illness and existing provisions and gaps. Recent work has been conducted in Sweden, Sri Lanka, and Malawi [5], as well as across Africa [3]. However, accurate estimates of the global burden of critical illness and the needs are not known. Some geographical areas have been less studied, and EECC research in the Americas, the Middle East, and Asia should be a priority, as should more granular research across different contexts at the national and local levels.

The implementing EECC questions include the determinants of why EECC is provided or not, the barriers to implementation and sustainability, indicators of adherence, and the core outcomes that should be measured when implementing EECC. Sustainable implementation will likely require developments in behavioural change as suggested by the theory of change models [16] that have been successfully used in other settings [17]. Research should investigate how such models and strategies can effectively implement EECC in various contexts and settings. Looking specifically at barriers, previous work on barriers to quality perioperative care delivery in LMIC identified four major themes: fragmented care pathways, limited human and structural resources, the costs of care for patients and patients' low expectations of care [18]. EECC can address these issues by shifting the focus away from siloed, fragmented care, emphasising the essential

**Table 1. Demographics of conference participants.**

| Gender | Male | 20 (43%) |
|---|---|---|
| | Female | 26 (57%) |
| Occupation | Physician | 30 (65%) |
| | Nurse | 2 (4%) |
| | Researcher (Exclusively) | 9 (20%) |
| | Technical Advisor | 2 (4%) |
| | Policy Maker | 3 (7%) |
| Work Settings | HICs | 23 (50%) |
| | LMICs | 8 (17%) |
| | Combination | 15 (33%) |

**Table 2. The top seven priority questions from each theme.**

**Current State**

| | |
|---|---|
| 1. | What is the global burden of critical illness? |
| 2. | How might conflict, climate catastrophes, pandemics and public health emergencies impact the critical illness burden and the burden on the health system? |
| 3. | What are the best definitions and criteria to measure critical illness? |
| 4. | What are the perceptions about critical illness, critical care, and EECC among different groups (patients, caregivers, health workers, hospital management, health authorities, policy-makers, global stakeholders)? |
| 5. | What are the ethical considerations, norms and practices around care for critically ill patients? |
| 6. | What are the current provisions for critical care, the existing gaps, prevailing clinical practices, knowledge levels among healthcare providers, and the overall readiness of health facilities to deliver essential critical care services? |
| 7. | In selected settings, what is the political economy of EECC (governance, finance, interest groups)? |

**Implementing EECC**

| | |
|---|---|
| 1. | What are the underlying determinants of whether EECC is provided or not? |
| 2. | What is the perceived value (cost, appropriateness, acceptability, feasibility) of EECC for stakeholders across different levels? |
| 3. | What are the key indicators of EECC adherence? |
| 4. | What are the facilitators and barriers for healthcare workers to implement and sustain EECC? |
| 5. | How can the components of EECC be operationalised in current policies, guidelines and curricula in different contexts? |
| 6. | What are the core outcomes we should measure in EECC research? |
| 7. | What can we learn from implementing other EECC-related initiatives, implementation models and quality improvement science? |

**Impact of EECC**

| | |
|---|---|
| 1. | What are the impacts of EECC on short and long-term mortality and morbidity? |
| 2. | What is the impact of EECC on outcomes in different patient subgroups? |
| 3. | What is the cost, cost-effectiveness, and budget impact of EECC? |
| 4. | What is the impact of implementing EECC on reducing preventable in-hospital mortality among critically ill patients? |
| 5. | What is the sustainability of EECC implementation? |
| 6. | How does EECC implementation affect patient flows through the health systems? |
| 7. | What are the broader impacts of implementing EECC, including unintentional consequences, for patients, providers, health systems, communities and the environment? |

**Refining EECC**

| | |
|---|---|
| 1. | Should the wording in Essential Emergency and Critical Care be changed to promote greater universal understanding of the concept? |
| 2. | Is a revision of the EECC package necessary at this stage, or should it be scheduled for a future date, and if so, when should this be? |
| 3. | Can AI be used to extract medical records of treatments provided to critically ill patients to inform modifications to EECC? |
| 4. | Which of the 40 defined EECC treatments and actions have the greatest impact on mortality and morbidity, and which are most cost-effective? |
| 5. | Should a reduced EECC package be developed for resource-deficient settings, e.g., pre-hospital and community settings? |
| 6. | Should EECC be differentiated for different population age groups, i.e., neonatal, paediatric and adult? |
| 7. | How best can foundational EECC be linked to more advanced critical care and definitive care programmes? |

**Table 3. The top ten priority questions.**

| Research Questions | Research Theme |
|---|---|
| What are the underlying determinants of whether EECC is provided or not? | Implementing EECC |
| What are the impacts of EECC on short and long-term mortality and morbidity? | Impact of EECC |
| What is the global burden of critical illness? | Current state |
| What are the facilitators and barriers for healthcare workers to implement and sustain EECC? | Impact of EECC |
| What is the impact of implementing EECC on reducing preventable in-hospital mortality among critically ill patients? | Impact of EECC |
| What are the key indicators of EECC adherence? | Implementing EECC |
| What are the current provisions for critical care, the existing gaps, prevailing clinical practices, knowledge levels among healthcare providers, and the overall readiness of health facilities to deliver essential critical care services? | Current state |
| What are the core outcomes we should measure in EECC research? | Implementing EECC |
| What is the impact of EECC on outcomes in different patient subgroups? | Impact of EECC |
| What is the cost, cost-effectiveness, and budget impact of EECC? | Impact of EECC |

resources required, and advocating for patients' understanding of and right to universal foundational care in critical illness, regardless of the underlying diagnosis.

The successful provision of EECC also depends on the availability of appropriate and context-sensitive training for healthcare providers. An EECC training course [19] has recently been developed, and other important training programs aim to improve emergency and critical care, such as the WHO's Basic Emergency Care (BEC) [20]. Further investigation is warranted to determine how these educational initiatives can be sustainably introduced across varied settings and to assess their impact on provider competencies and patient outcomes. Ultimately, the long-term provision of EECC will require integration with WHO and other stakeholder initiatives, and inclusion in countries' health systems through a coordinated approach that involves multiple implementation strategies alongside training, including policy development, leadership capacity building, and data collection and evaluation mechanisms to ensure its effectiveness and sustainability across various health system contexts [21].

Key questions regarding the impact of EECC include its influence on both short- and long-term health outcomes, the proportion of mortality potentially preventable through its implementation, its differential effects across patient subgroups, and its cost-effectiveness and budget implications. Generating robust evidence in these areas is crucial for informing policy decisions and allocating resources effectively. While research into effective implementation strategies is necessary, equal attention must be given to evaluating the impact of EECC. This will require identifying appropriate outcome measures and developing rigorous methodological approaches to assess their effectiveness and value across diverse healthcare settings. These goals align with the World Health Assembly Resolution WHA75.8, which establishes a global mandate to strengthen the quality of clinical trials, provide high-quality evidence on health interventions and improve research coordination [22]. The resolution aligns strongly with the need to generate robust, policy-relevant evidence on the effectiveness of EECC implementation, design studies that reflect diverse health systems, and build multidisciplinary networks.

Questions related to the refinement of EECC were not prioritised among conference participants. While this may reflect broad acceptance of the current EECC framework, it is also important to acknowledge the potential influence of selection bias. Conference participants were purposively invited based on their expertise and engagement with EECC, and many

were already familiar with, or invested in, the existing framework. This may have limited the extent to which more fundamental or critical refinement questions were proposed or prioritised. As EECC continues to evolve, future priority-setting exercises that include a broader range of stakeholders and perspectives, particularly those less closely involved in the development of the current framework, may place greater emphasis on refinement. In addition, emerging scientific evidence and technological developments, including advances in artificial intelligence, may further prompt reassessment and adaptation of EECC.

There are several strengths to the methodology used in this research priority-setting process that enhance the validity of the questions generated. The use of a structured, stepwise framework, adapted from the established approach described by Schmidt [8], enabled a systematic progression from question generation to final prioritisation. This design ensured that all participants had multiple opportunities to engage meaningfully throughout the process. The combination of small-group work and plenary sessions promoted collaborative reflection while supporting in-depth, theme-specific exploration. The use of anonymous digital tools (Menti and SmartSurvey) was employed to encourage open and honest input, thereby attempting to mitigate the influence of power dynamics and hierarchical bias, particularly during the ranking phase. A further strength was the inclusion of participants from a range of geographic regions and diverse clinical and research backgrounds, which brought varied contextual perspectives and increased the global applicability of the resulting priorities. Finally, circulating the draft manuscript for participant feedback added a further layer of validation and transparency to the process.

There were also limitations. While the participant group was diverse, encompassing clinical, academic, and policy expertise, it lacked broader representation from other key stakeholder groups, including health system managers, funders, nurses, and patient or community representatives. This limited the scope of perspectives, particularly regarding system-level integration and community impact. Specifically, for patient and community representatives, the conference was structured as a highly technical forum focused on research methodology, clinical priorities, and health system considerations. Participation was therefore limited to researchers and clinicians with relevant expertise to enable detailed discussion within a constrained timeframe. This decision was not intended to undervalue patient or community perspectives, which are essential to the relevance, equity, and acceptability of EECC. Since the conference, we have strengthened engagement with patient and community groups within the broader EECC programme, and their meaningful involvement will be central to future research activities. The geographical scope was also relatively narrow. Although participants represented varied economic and global contexts, only nine largely high-income countries were included, which may limit the diversity of viewpoints.

Reflection should also be given to several methodological considerations that may have influenced the prioritisation outcomes. First, although experienced facilitators were used to support inclusive discussion, we did not employ a formal method to observe or measure dominance patterns during the in-person group discussions. As such, it is possible that more vocal or senior participants exerted greater influence during the early stages of idea generation, particularly before the introduction of anonymous ranking procedures. Future priority-setting initiatives could mitigate these risks by incorporating more structured facilitation techniques that explicitly encourage dissent, ensuring greater diversity in participant selection, and providing multiple modes of engagement (e.g., anonymous inputs, asynchronous contributions) to reduce the influence of hierarchy and enhance inclusivity Second, the compressed two-day conference format, while necessary for feasibility, may have limited opportunities for extended reflection and deeper deliberation on complex or cross-cutting research questions. Time constraints may have favoured more readily articulated or familiar topics over emerging or conceptually challenging issues. Third, the thematic grouping of participants, although useful for structuring discussion and ensuring coverage of key domains, may have contributed to silo effects. Some research questions, particularly those spanning implementation, outcomes, and health systems, may have been constrained by thematic boundaries, potentially limiting cross-thematic synthesis or integration. Furthermore, although the final ranking process effectively captured the perceived importance of each research question; other critical prioritisation

criteria, such as feasibility, equity, urgency, and expected impact, were not systematically integrated into the scoring framework. These limitations reflect the inherent trade-offs between feasibility, inclusivity, and depth in face-to-face priority-setting processes. Future agenda-setting efforts may benefit from extended timelines, iterative engagement, and complementary methods, such as pre-conference surveys or subsequent Delphi rounds, to strengthen deliberation and integration.

## Conclusion

Following the first global research conference on Essential Emergency and Critical Care, a set of ten priority research questions has been identified to guide the EECC research agenda over the next five years. These priority questions highlight gaps in knowledge regarding current understanding of burden and care provision, implementation challenges, and the impact of EECC on health outcomes and systems. The next step will be to engage a broad and diverse constituency of stakeholders, including nurses, frontline clinicians, researchers, health policymakers, patient and community advocates, governments, and funders, to advance these priorities collaboratively, while ensuring that future priority-setting processes incorporate explicit considerations of equity, feasibility, and urgency alongside technical importance.

## Supporting information

**S1 File. Expert participants.**
(DOCX)

**S1 Data. Data.**
(DOCX)

## Acknowledgments

**The EECC Prioritisation Group members**: Karima Khalid, Dan Brun Petersen, Save Schroder, Kun Arifi Abbas, Septo Sulistio, Alhassan Datti Mohammed, Andreas Barratt-Due, Nobhojit Roy, Andreas Wellhagen, Birger Forsberg, Carina King, Celia Blaas, Christoffer Hintze, Claudia Hanson, Emily Tegnell, Helle Mølsted Alvesson, Jonas Blixt, Louise Elander, Mariam Claeson, Miklos Lipcsey, Petronella Bjurling-Sjöberg, Tobias Alfvén, Aneth Charles Kaliza, Anna Hvarfner, Godfrey Barabona, Ulrika Baker, Nick Leech, Teresa Bleakly Kortz, Halinder Mangat, Jonna idh, and Simon-Fredrik Schell.

## Author contributions

**Conceptualization:** Alexandra Wharton Smith, Martin Gerdin Wärnberg, Carl Otto Schell, Tim Baker.

**Data curation:** Ryan Rhys Ellis.

**Formal analysis:** Ryan Rhys Ellis, Alexandra Wharton Smith, Martin Gerdin Wärnberg.

**Investigation:** Tim Baker.

**Methodology:** Alexandra Wharton Smith, Carl Otto Schell, Tim Baker.

**Project administration:** Alexandra Wharton Smith, Martin Gerdin Wärnberg, Carl Otto Schell, Tim Baker.

**Resources:** Tim Baker.

**Supervision:** Tim Baker.

**Writing – original draft:** Ryan Rhys Ellis, Wurry Ayuningtyas.

**Writing – review & editing:** Ryan Rhys Ellis, Wurry Ayuningtyas, Alexandra Wharton Smith, Martin Gerdin Wärnberg, Carl Otto Schell, Tim Baker.

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
