## [Decision Letter · Decision Letter 0]

28 Oct 2025

PGPH-D-25-02550

Top ten research priorities for Essential Emergency and Critical Care: A modified Delphi process

Dear Dr. Baker,

Thank you for submitting your manuscript to PLOS Global Public Health. After careful consideration, we feel that it has merit but does not fully meet PLOS Global Public Health’s publication criteria as it currently stands. Therefore, we invite you to submit a revised version of the manuscript that addresses the points raised during the review process.

We look forward to receiving your revised manuscript.

Kind regards,

Vinay Nair Kampalath, MD, MSc, DTM&H

Academic Editor

Journal Requirements:

1. Please send a completed 'Competing Interests' statement, including any COIs declared by your co-authors. If you have no competing interests to declare, please state "The authors have declared that no competing interests exist". Otherwise please declare all competing interests beginning with the statement "I have read the journal's policy and the authors of this manuscript have the following competing interests:"

Additional Editor Comments (if provided):

Thanks for this excellent contribution.

If you can submit a minor revision with attention to Reviewer 1's comments/suggestions, that would be appreciated and sufficient.

Reviewers' comments:

Reviewer's Responses to Questions

**Comments to the Author**

1. Does this manuscript meet PLOS Global Public Health’s publication criteria?

Reviewer #1: Yes

Reviewer #2: Yes

Reviewer #3: Yes

2. Has the statistical analysis been performed appropriately and rigorously?

Reviewer #1: N/A

Reviewer #2: Yes

Reviewer #3: N/A

3. Have the authors made all data underlying the findings in their manuscript fully available (please refer to the Data Availability Statement at the start of the manuscript PDF file)?

Reviewer #1: Yes

Reviewer #2: No

Reviewer #3: Yes

4. Is the manuscript presented in an intelligible fashion and written in standard English?

Reviewer #1: Yes

Reviewer #2: Yes

Reviewer #3: Yes

Reviewer #1: Specific Recommendations & Line-Level Suggestions

Introduction

The background on the burden of critical illness is thorough and presents a compelling rationale. However, the manuscript would benefit from one or two sentences clarifying why previous efforts at priority setting or global critical care research did not yield a similar list specifically, what exactly is new? and why now? - what motivated the timing and structure of this initiative? (Consider expanding lines 79–82.)

For non-specialist readers, it would be helpful further clarify EECC from other advanced critical care definitions -what does EECC include and exclude? (Lines 70–78.)

Methods

Specifically describe how thematic groups were assigned to their focus and if any attempt was made to balance expertise, gender, or regional background across groups (Lines 96–100).

Clarify whether all 46 participants were involved in every round of each process, or if attrition occurred between sessions (Lines 106–131). Reporting actual completion rates provides more credibility and transparency.

When describing the anonymous polling and surveys, briefly mention the rationale for tool choices (e.g., why SmartSurvey vs. other platforms) and any technical issues or accommodations.

Results

In Table 1, adding percentages next to raw numbers improves quick comprehension for an international audience.

For rated questions, please consider including standard deviations, interquartile ranges, or a brief commentary on which questions were most (or least) universally endorsed versus more contentious.

In discussing final priorities (Table 3), briefly explain why no top-ranked questions emerged from the “refining” theme and was this an unexpected outcome or does it reflect broad agreement on existing definitions?

Discussion

The discussion is appropriately balanced. Still, more time could be spent on the limitations of the process. For example: the optimism in participant diversity is acknowledged, but clearly noting that only a handful of countries were represented, and no patient/community voices were involved, is crucial. First, consider a sentence quantifying how many countries were represented and explicitly acknowledging the absence of these key perspectives (Lines 229–236).

Consider expanding the discussion (Lines 237–244) on how inclusion/exclusion criteria, group facilitation, or digital tools may have inadvertently shaped (or limited) participation/dissent due to hierarchical or dominant voices. A reflective comment on how this could be mitigated in future efforts would help.

When proposing next steps, add concrete suggestions for broadening engagement (nurses, health policymakers, patient advocates) and for integrating equity/feasibility/urgency into future rankings, beyond technical importance alone.

Stylistic and Minor Points

Lines 81–82: “In this paper, we report the output of an exercise…”—specify “a consensus-driven exercise to generate global EECC research priorities for the next five years.”

Lines 154–159: Consider reordering the “current state” questions to emphasize the global burden question first, followed by definitions and then system-level gaps/barriers.

Make a final proofread for any typographical or grammar inconsistencies. The overall English is very strong.

Data, Ethical, and Competing Interests

The ethical explanation supporting the absence of formal review is valid for this kind of expert priority-setting, but adding a brief reference to other consensus exercises (in global health or health policy) that followed similar standards could be assuring for editors and readers less familiar with Delphi approaches.

Data Availability

If possible, upload as supplementary materials or via a repository the full list of all 28 questions, with raw (de-identified) scores from each participant, to maximize transparency and utility for secondary analysis.

Reviewer #2: Title:

Clear

Abstract:

What does EECC describe? Is it a short course you are going to offer? Is it equipment? is it a protocol? Not clear

Financial Disclosure:

Done

Competing Interest:

DOne

Data availability:

Full responses before selection need to be provided.

Introduction:

-Problem is clear mode of delivery of EECC is not clear as mentioned? Is it a course, a set of protocols? Define.

-What is the current model of EECC? (referring to line 212), would be great to highlight from the beginning.

Study design?

-What was the total number of participants in the conference? Was 46 all or did you do selection?

-What was the selection criteria? Not clear

-Define relevant clinical backgrounds? Do you mean relevant to emergency and critical care?

-Define research interests? Emergency and critical care related or others?

Participants:

-You mention that participants were purposely assigned. What was the distribution in terms of expertise among the groups? Was it homogenous or heterogenous group? Not clear.

-Explicitly mention the size of each group.

Prioritization process:

-Online document for analysis? Should be provided in the annexes for review.

-Who decided that 7 questions were to be picked? Was it research team leads or participants or both? Why 7? Why not any other number?

-Closing criteria was demonstrated and defined prior.

Ethics:

-Written informed Consent was obtained from participants which is great.

-Survey responses were anonymous, good.

-Mention whether the online document used for analysis was anonymous as well.Provide this as annex too

Results:

-Well presented

-Provide all the generated questions as supplementary data.

Discussion:

-Good.

-Strengths and limitations are highlighted.

-You have mentioned the 2 days of conference as limitation, should you have considered extension of time using electronic media to get extensive input?

-Why did you not do the actual Delphi technique?

Reviewer #3: The authors aimed to provide direction in the area of research into essential emergency and critical care (EECC) by assessing and prioritizing research questions for future academic enquiry. This was done using a Delphi method where a panel of experts provided consensus opinions on major areas in EECC research, and through sequential polling, prioritized ten research questions. Out of four initially proposed subthemes, three subthemes areas agreed upon as having top priority for further research. These were the current state of EECC, the implementation of EECC, and the impact of EECC. The chosen method of the Delphi study was appropriate to answer the question as it not only made the study feasible, it also provided a scientific way of providing answers to a complex research question.

Overall, the study was well designed, and the article well written, and does provide sufficient information on the proposed question. Limitations which were inherent to the study design and other aspects of the methodology were duly acknowledged by authors, and attempts were made to compensate for them appropriately, improving the robust nature of the study design. The importance of driving the research agenda for EECC is made apparent by authors throughout the article, with clear and concise academic language, without being dense or difficult for a reader with limited technical expertise to appreciate.

If published, the study would not only increase interest in the field of EECC, it would also serve as a springboard for further research and academic enquiry. It is my recommendation that this article be accepted, with some minor areas for improvement or revision.

While the authors do provide some background to the study, a more in-depth review of existing literature would be appreciated, to help frame the study’s relevance and provide more background as to why the study is important.

The limited scope of the expert panel in terms of areas of practice as well as geographical distribution does pose a limitation to the strength and validity of the final research questions generated. Further details on the qualifications of the expert panel could be provided, such as their clinical areas, and scope of practice, to help improve the merit of the panel’s recommendations. The strength of the Delphi method used is largely dependent on the quality of the experts assembled, and more details about the panel would be useful in increasing the strengths of the study. This can be included as supplemental data, and does not necessarily need to be included in the text of the article.

**Do you want your identity to be public for this peer review?** For information about this choice, including consent withdrawal, please see our Privacy Policy

Reviewer #1: No

Reviewer #2: No

Reviewer #3: **Yes:** Nanaba Adoma Dawson-Amoah

---

## [Decision Letter · Decision Letter 1]

16 Dec 2025

PGPH-D-25-02550R1

Top ten research priorities for Essential Emergency and Critical Care: A modified Delphi process

Dear Dr. Baker,

Thank you for submitting your manuscript to PLOS Global Public Health. After careful consideration, we feel that it has merit but does not fully meet PLOS Global Public Health’s publication criteria as it currently stands. Therefore, we invite you to submit a revised version of the manuscript that addresses the points raised during the review process.

The manuscript has been evaluated by two reviewers, and their comments are available below.

The reviewers have raised a number of remaining concerns. These include concerns over methodology and data transparency, as well as providing further clarity regarding the reporting of participants.

Could you please carefully revise the manuscript to address all comments raised?

We look forward to receiving your revised manuscript.

Kind regards,

Jen Edwards

Staff Editor

Journal Requirements:

1. Please amend your online Financial Disclosure statement. If you did not receive any funding for this study, please simply state: “The authors received no specific funding for this work.”

2. Please send a completed 'Competing Interests' statement, including any COIs declared by your co-authors. If you have no competing interests to declare, please state “The authors have declared that no competing interests exist". Otherwise please declare all competing interests beginning with the statement "I have read the journal's policy and the authors of this manuscript have the following competing interests:

For more information, please go to our submission guidelines:

https://journals.plos.org/globalpublichealth/s/submission-guidelines#loc-competing-interests

Additional Editor Comments (if provided):

Reviewers' comments:

Reviewer's Responses to Questions

**Comments to the Author**

Reviewer #1: All comments have been addressed

Reviewer #2: All comments have been addressed

publication criteria?

Reviewer #1: Yes

Reviewer #2: Yes

3. Has the statistical analysis been performed appropriately and rigorously?

Reviewer #1: Yes

Reviewer #2: Yes

4. Have the authors made all data underlying the findings in their manuscript fully available (please refer to the Data Availability Statement at the start of the manuscript PDF file)?

Reviewer #1: Yes

Reviewer #2: Yes

5. Is the manuscript presented in an intelligible fashion and written in standard English?

Reviewer #1: Yes

Reviewer #2: Yes

Reviewer #1: The authors have significantly improved this manuscript. The remaining issues are primarily focused on data transparency and critical self-reflection on the methods. The work is valuable and timely, but to be acceptable for publication, it must meet the journal's standards for data sharing. I recommend conditional acceptance upon the satisfactory completion of the mandatory changes, particularly the provision of the supplementary quantitative data table.

Reviewer #2: Abstract and Introduction

“What does EECC describe? Is it a course, equipment, protocol?”

We expanded the descriptions in the introduction and main text, clarifying that EECC

comprises clinical processes, readiness requirements, tools, and training materials.

Comment: Response is satisfactory

“What is the current model of EECC?”

This has been clarified and introduced earlier in the manuscript.

Comment: now clear

Study Design and Participants

“Total number of conference participants? Was 46 all or selected?”

We added details explaining total attendance, inclusion criteria, and selection of the 46

participants.

Comment: Not clear whether total attendees were 46 or more. State this explicitly if this was the case. You mention vast majority completed. What number out of 46 did actually complete. Be explicit (line 105-107)

“Define relevant clinical backgrounds and research interests.”

We clarified that backgrounds and interests were relevant to emergency and critical care,

and have added a supplementary appendix detailing all expert backgrounds (Supplemental

Material 1).

Comment: Now clear. Supplementary material satisfactory. I however note that list has 32 participants and not 46. Is there an explanation for this?

“Distribution of expertise across groups—heterogeneous or homogeneous?”

We added a sentence describing the deliberate distribution of expertise to achieve balanced

groups.

Comment: Noted. If the distribution of the cohorts is available, it would be good to display as supplementary material. If not available, can be accepted as is.

Prioritisation Process

“Clarify the ‘online document’.”

We clarified that SmartSurvey was used for data collection and that all responses were

anonymous.

Comment: response satisfactory.

“Who decided on selecting seven questions per theme?”

A clarifying sentence has been added, noting that research leads proposed seven per group

to maintain equal representation across themes.

Comment: Noted but not clear why the number 7 was chosen by research leads. Is it based on some prior research? What is the basis for 7?

Ethics

“Was the online document anonymous?”

Yes — this is now stated explicitly.

Comment: response satisfactory

Discussion

“Consider extension via electronic media; why not a full Delphi?”

We added a statement noting that time constraints prevented a multi-round Delphi, but that

future reviews could incorporate extended electronic input.

Comment: response satisfactory

**Do you want your identity to be public for this peer review?** For information about this choice, including consent withdrawal, please see our Privacy Policy

Reviewer #1: No

Reviewer #2: No

---

## [Decision Letter · Decision Letter 2]

12 Feb 2026

Top ten research priorities for Essential Emergency and Critical Care: A modified Delphi process

PGPH-D-25-02550R2

Dear Dr Baker,

We are pleased to inform you that your manuscript 'Top ten research priorities for Essential Emergency and Critical Care: A modified Delphi process' has been provisionally accepted for publication in PLOS Global Public Health.

Best regards,

Julia Robinson

Executive Editor

Reviewer Comments (if any, and for reference):

Reviewer's Responses to Questions

**Comments to the Author**

Reviewer #2: All comments have been addressed

publication criteria?

Reviewer #2: Yes

3. Has the statistical analysis been performed appropriately and rigorously?

Reviewer #2: Yes

4. Have the authors made all data underlying the findings in their manuscript fully available (please refer to the Data Availability Statement at the start of the manuscript PDF file)?

Reviewer #2: Yes

5. Is the manuscript presented in an intelligible fashion and written in standard English?

Reviewer #2: Yes

Reviewer #2: The responses have addressed all the concerns raised and provides clear methodology to the reader. Limitations have been well expounded as well and serves as baseline for future research in EECC

**Do you want your identity to be public for this peer review?** For information about this choice, including consent withdrawal, please see our Privacy Policy

Reviewer #2: No
